# Sequence of Rare Diagnoses in a Young Patient: Altitude Barotrauma Hemopneumothorax and Desquamative Interstitial Pneumonia

**DOI:** 10.3390/diagnostics13142367

**Published:** 2023-07-14

**Authors:** Iustina Leonte, Karina Ivanov, Angela Ștefania Marghescu, Șerban Radu Matache, Florica Valeria Negru, Ana Luiza Iorga, Silviu Mihail Dumitru, Beatrice Mahler

**Affiliations:** 1Institute of Pneumology “Marius Nasta”, 050159 Bucharest, Romania; stefania_angela@yahoo.com (A.Ș.M.); radu.matache@gmail.com (Ș.R.M.); dr.valer.n81@gmail.com (F.V.N.); ana-luiza.iorga@drd.umfcd.ro (A.L.I.); silviu.dumitru@marius-nasta.ro (S.M.D.); beatrice.mahler@umfcd.ro (B.M.); 2Faculty of Medicine, “Carol Davila” University of Medicine and Pharmacy, 050474 Bucharest, Romania

**Keywords:** hemopneumothorax, altitude barotrauma, emphysematous lesions, desquamative interstitial pneumonia

## Abstract

We present the case of a 35-year-old patient without pathological history who developed hemopneumothorax due to altitude barotrauma during a commercial airline flight. The computed tomography (CT) of the chest identified the presence of right hydropneumothorax and emphysema “blebs” and bubbles. After the therapeutic insertion of a drain tube, the patient returned to the country by land transport. Three weeks later, he was diagnosed with right-sided pleurisy based on a CT scan with contrast material. A surgical intervention was then performed, and three biopsy samples were taken; the histopathological result highlighted suggestive elements for the diagnosis of desquamative interstitial pneumonia (DIP).

**Figure 1 diagnostics-13-02367-f001:**
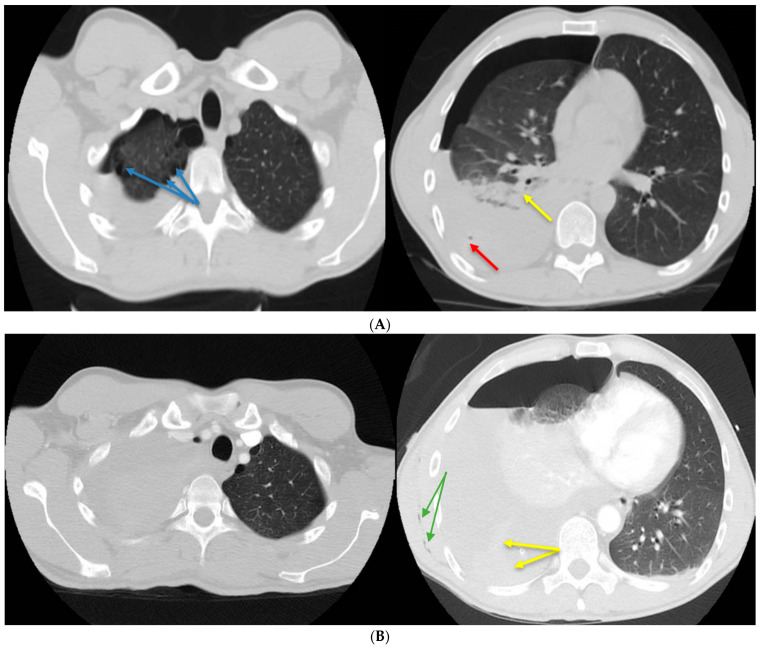
(**A**) Native chest computed tomography (CT). Moderate right hemopneumothorax with the maximum thickness in the anteroposterior diameter of the liquid component of approximately 4.5 cm and the air component of approximately 2 cm, with air bubbles included (red arrow), which associates secondary partial passive atelectasis (yellow arrow) of the right lung. Bilateral subpleural emphysema bubbles between 8 and 25 mm in size (blue arrows). (**B**) Chest CT with contrast material. Basal visible pleural drain tube on the right side. Quantitative increase in the right hemopneumothorax, with significant passive atelectasis of the entire lung and displacement of the contralateral mediastinal structures (criteria of severity). Adjacent to the distal end of the drain tube, spontaneous hyperdensities (blood clots) can be noted in the fluid component. Minimal subcutaneous emphysema is visible along the latissimus dorsi muscle (green arrows). A 35-year-old patient, a 10-pack/year smoker with no history of chronic pathologies, developed sudden rest dyspnea during an air flight, paresthesias in the right hemibody, and intense pain in the right hemithorax. After landing, he used an emergency service where a brain CT (computed tomography) was performed, excluding possible neurological lesions. The chest CT revealed a moderate right hydropneumothorax, with both gas bubbles and bilateral subpleural emphysema bubbles, with sizes between 8 and 25 mm (Figure 1A). A drain tube was installed to evacuate the hydropneumothorax, but in the next 24 h, the patient developed anemia, with a drop in hemoglobin from 14 to 8.1 g/dL. In the absence of clinical improvement and paraclinical degradation, it was decided to transfer the patient to another hospital unit. Despite pleural drainage, the hydropneumothorax persisted. Due to persistent hydropneumothorax, another chest CT scan (Figure 1B) was performed. A higher-caliber drain tube of 28 Ch was placed for three days, and approximately liters of bloody pleural fluid was evacuated with the biochemical characteristics of an exudate (proteins = 3.4 g/dL). Afterward, the evolution was favorable with the remission of the hemopneumothorax, re-expansion of the right lung, an increase in hemoglobin values, and a major clinical improvement. All of these led to the discharge of the patient and his return to the country by land transport.

**Figure 2 diagnostics-13-02367-f002:**
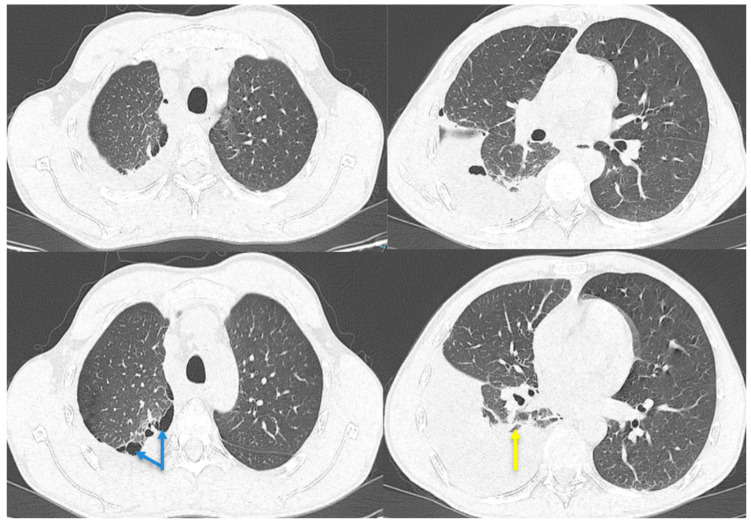
Chest CT with contrast agent. Pleural effusion on the right side closed apically and basally, with several air inclusions up to 4.3 cm thick, which associate with underlying pulmonary compression (yellow arrow); pulmonary consolidation involving quasi-complete LID with an atelectatic component; paraseptal emphysema in the upper lobes (blue arrows). After approximately 3 weeks, the patient went to the emergency room once more, complaining of exertional dyspnea, dry cough, pain in the right hemithorax, febrile syndrome with a maximum T value of 38.8 °C, and a weight loss of approximately 5 kg in 3 weeks. At the time of presentation, the patient was afebrile, normotensive, with oxygen saturation within normal limits. The clinical examination revealed the existence of an asthenic chest, with the absence of the transmission of vocal vibrations in the lower third of the right hemithorax, dull on percussion, and a diminished vesicular murmur on lung auscultation in this area. The blood tests revealed the existence of mild leukocytosis with neutrophilia, mild thrombocytosis, moderate anemia, and inflammatory syndrome. Chest CT with a contrast agent was performed, highlighting right pleural effusion with air inclusions, almost complete atelectasis of the right lower lobe, and paraseptal emphysema in the upper lobes (Figure 2).

**Figure 3 diagnostics-13-02367-f003:**
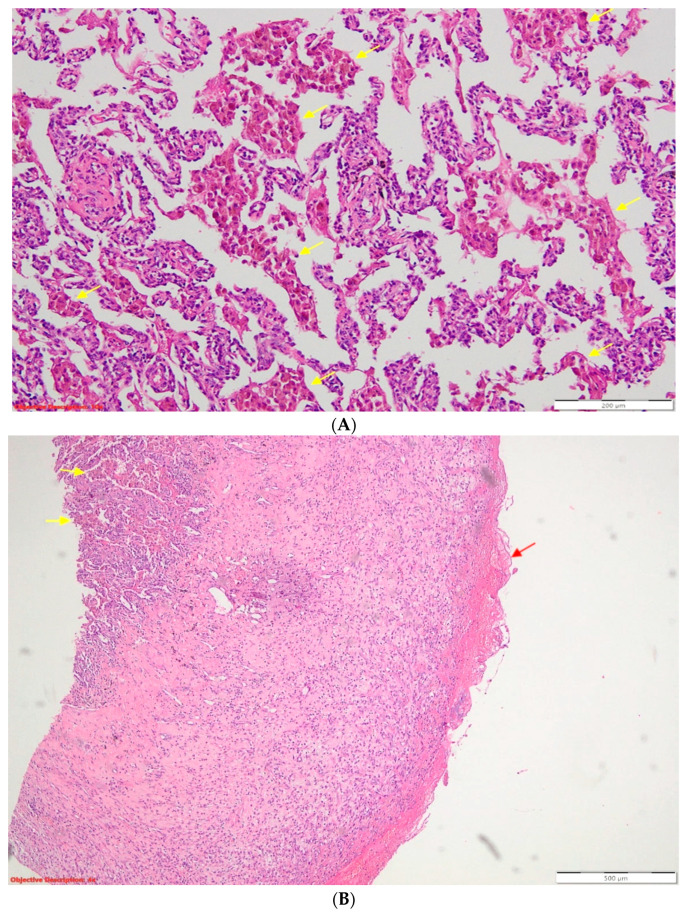
(**A**) Desquamative pneumonia (DIP-like) reaction—endoalveolar collections of macrophages with intracytoplasmic brown pigment are present (yellow arrows); HE, 100×; HE = standard stain (hematoxylin–eosin). (**B**) Fibrinous pleurisy in the process of organization—the visceral pleura is thickened, and on its surface, there are eosinophilic deposits of fibrin (red arrow). The underlying lung parenchyma presents numerous macrophages with intracytoplasmic brown pigment (yellow arrows) inside the alveolar spaces. Desquamative pneumonia—DIP-like reaction; HE, 40×. Thoracentesis was performed for diagnostic purposes with the extraction of 40 mL of hemorrhagic fluid, and the biochemical examination revealed the characteristics of an exudate (proteins = 3.6 g/dL). The cytological examination identified frequent lymphocytes, relatively frequent macrophages, and mesothelial cells. From the sediment of the pleural fluid, a cytologic was performed, which was examined on four serial sections consisting of rare lymphocytes intermixed with neutrophils. Antibiotherapy with third-generation cephalosporines, non-steroidian anti-inflammatory, antitussive, and analgestic treatment was initiated. Afterward, the patient was transferred to the thoracic surgery department, where exploratory and curative thoracoscopy were performed: the identified clot was evacuated, the pneumothorax was surgically cured, and fragments of lung tissue were taken for histopathological examination (HP). The result of the HP examination described histopathological aspects compatible with the diagnosis of pleural emphysema, fibrinous pleuritis in the process of organization, and elements suggestive of the diagnosis of DIP: abundant macrophages with intracytoplasmic brown pigment located endoalveolarly and in the bronchiolar endoluminal space (Figure 3).

**Figure 4 diagnostics-13-02367-f004:**
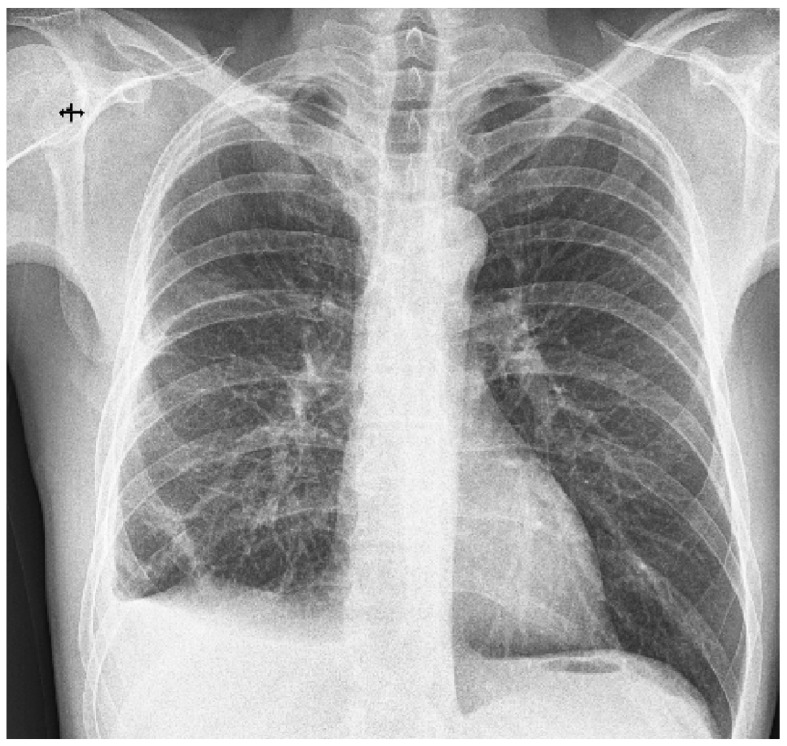
Chest X-ray 1 week post-operatively. Small amount of pleural fluid at the level of the right costo-diaphragmatic sinus; normal expansion of the right lung. Remaining fibroatelectatic bands in the lower third of the right lower lobe. The post-operative evolution was favorable; on the chest X-ray taken one week after the intervention, the normal expansion of the lung was observed after six days of continuous drainage on a 28 Ch tube (Figure 4). The patient was discharged with the recommendation to avoid travel at altitude, abandon smoking, and undergo a CT-imaging evaluation 3 months after surgery. Barotrauma sums up the tissue damage that occurs as a result of pressure differences between the internal and external environment. This occurs when the body is exposed to a change in the pressure of the external environment, as happens in the case of divers or aircraft pilots [1]. Along with the formation of massive, suddenly installed pneumothorax, hemothorax can also be associated, with the source of hemorrhage being the arteries that vascularize the visceral pleura or those located at the level of the pulmonary parenchyma subjected to the mechanical stress of bursting emphysema lesions. The primary type of pneumothorax most commonly affects young age groups, 20–30 years, with a seven times higher incidence among men than women and is most commonly caused by rupture of asymptomatic blebs/bullae emphysematous lesions, identified in 80% of cases of spontaneous pneumothorax. It is also associated with other risk factors, such as a low body mass index, a longilin, asthenic constitutional type, Marfan syndrome, pregnancy, and smoking [2]. Chronic smoking is the main risk factor encountered in desquamative interstitial pneumonia (DIP), being associated with it in more than 90% of cases [3]. The diagnosis of DIP is based on clinical and radiological criteria to which, ideally, the histopathological criteria are added. On high-resolution computer tomography, the pathognomonic lesions described are diffuse ground-glass opacities arranged predominantly basally, peripherally, and symmetrically, accompanied by the presence of small air cysts, with this association of lesions being found in approximately one-third of patients. Alveolar septal thickening and centrolobular emphysema can also be found [4,5]. Histopathologically, DIP is characterized by the presence of macrophages with intracytoplasmic brown pigment intraalveolarly located. Multinucleated giant cells, alveolar wall thickening, and sporadic eosinophils may also occur [6]. In the presented case, chronic exposure to cigarette smoke created the fertile ground for the development of emphysematous lesions, with the formation, over time, of emphysema blebs and bubbles that, at atmospheric pressure differences, can cause spontaneous pneumothorax [2]. Smoking is also an important risk factor in DIP, according to the studies published so far [7]. It is noteworthy that, in addition to chronic exposure to cigarette smoke, our patient also meets other predisposing conditions for the installation of pneumothorax, such as young age, male sex, longilin constitutional type, asthenic chest, and low body mass index. This association of factors makes pneumothorax possible through altitude barotrauma, and the presence of blebs-bullae, emphysematous lesions increases the risk of pneumothorax recurrence. A retrospective study of 115 patients with spontaneous pneumothorax cites a 39% recurrence rate, emphasizing a higher prevalence of smokers (57% vs. 22%) as well as the impact of lesions of respiratory bronchiolitis associated with smoking on the risk of repositioning pneumothorax [8,9]. The cascade of smoking–chronic inflammation–DIP–pneumothorax is the cornerstone in the correct management of the case. The target of treating such a patient is to avoid recurrences in terms of pneumothorax and limit the progression of DIP characteristic lesions while maintaining pulmonary function as close to physiological as possible. Some authors emphasize the importance of surgical cure of persistent emphysematous lesions after a first episode of spontaneous pneumothorax, given the increased risk of recurrence in their presence [10]. In our case, the restoration of the hemothorax two weeks after the initial drainage required a surgical cure via VATS (video-assisted thoracoscopic surgery) with the evacuation of the existing clot and the sampling of the pleuro-pulmonary fragments, whose histopathological examination provided the second revealing diagnosis: desquamative interstitial pneumonia. Long-term corticosteroid therapy has been reported to be most effective in the treatment of DIP [11]; however, considering the well-known risks and dealing with a young patient with adequate pulmonary reserves, the administration of corticosteroids was delayed. Moreover, the histopathologic aspect of DIP suggested an early stage disease that could not have been visualized on a CT scan yet. Therefore, it is recommended to eliminate the risk factors by giving up smoking, avoiding flights and exposure to large pressure differences, as well as establishing the available therapeutic measures for the rapid recovery of respiratory function: respiratory physiotherapy. Searching in the specialized literature in the databases PubMed, ClinicalKey, and Biefield Academic Search Engine for the keywords “high altitude pulmonary barotrauma and pulmonary interstitial disease”, and “high altitude pulmonary barotrauma and desquamative pneumonia”, we found two case report articles that associated the two pathologies [12]. These studies showed the association of spontaneous pneumothorax with DIP, but the diagnosis of pneumothorax occurred after the diagnosis of DIP. In none of the mentioned articles was the pneumothorax produced by altitude barotrauma.

## Data Availability

Not applicable.

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
