# Peer review of "Sequence of Rare Diagnoses in a Young Patient: Altitude Barotrauma Hemopneumothorax and Desquamative Interstitial Pneumonia"

_diagnostics, 2023, doi:10.3390/diagnostics13142367_

Round 1
Reviewer 1 Report
Dear authors,
I read your case regarding a young man with spontaneous PTX and DIP with interest.
-The manuscript lacks structure (such as introduction, case report, discussion, conclusion).
-Extensive english language is needed (ex past tense instead of present)
-Please describe whether the initial pneumothorax was large (>2cm, severe symptoms) to address why evacuation drainage without a leave-in tube was not attempeted. What were the sizes of the bores?
-What were the initial characteristics of the pleural fluid?
-On what etiology was the hemorrhage attributed? Please explain further on the discussion part, once formatted.
-Please describe if there was persistent air-leakage and after how many days was the chest tube removed. Was there complete expansion of the lung after removal?
-Line 60: what are the characteristics of the fluid? ex lymphocytic, PMN?
-Please correct the references (ex xiii is the same with xiv)
Overall Comment: Although this is an interesting case, extensive corrections of the lunguage, the structure and the discussion are needed, thus leading to rejection.
Best regards.
Language editing is needed
Author Response
Dear reviewer,
We made the suggested additions to our article, where possible, and we have explained them point by point in the attached document. Thank you for all your suggestions.
Please see the attachment.

Reviewer 2 Report
This report present the case of a 35-year-old patient without pathological history who developed hemopneumothorax due to altitude barotrauma during a commercial airline flight. Further histopathological examination of the lung fragment identified elements suggestive for the diagnosis of respiratory bronchiolitis associated with desquamative interstitial pneumonia (DIP). Overall, the case is interesting and the manuscript is well-written.
Minor suggestion
1. Please introduction to briefly discuss the presentation of DIP.
2. Please add the detailed result of pleural effusion - neutrophil or lymphocyte - predominate exudate.
3. What do you mean by anti-inflammatory agents at line 61? Is it corticosteroid?
Author Response
Dear reviewer,
We made the suggested additions to our article, where possible and we have explained them point by point in the attached document. Thank you for all your suggestions.
Please see the attachment.

Reviewer 3 Report
In this article, Leonte et al present the case of a 35-year-old patient without pathological history who developed hemopneumothorax due to altitude barotrauma during a commercial airline flight. The computed tomography of the chest identified the presence of emphysema "blebs" and bubbles, whilst the histopathological examination of the lung fragment identified elements suggestive for the diagnosis of respiratory bronchiolitis associated with desquamative interstitial pneumonia (DIP). It is a very valuable case, and I think it is a very useful case for clinicians.
major concerns)
1) In Figures 1 and 2, the CT is in the stage of hemothorax now, so it is difficult to determine whether DIP is really present or whether it is reactive to hemothorax in a one-point view. Therefore, even after the hemothorax has healed, it may be necessary to reconfirm whether there are interstitial shadows that may be suspicious for DIP. If you have CT images, please provide them.
minor concerns)
1) Please confirm whether Roman numerals are acceptable for the literature number, as well as the submission rules.
Author Response

(The authors gave the same response as above.)
